# What Does a KAP Survey Reveal about the Awareness Regarding Leishmaniasis among the Community of an Endemic Area in Sri Lanka?

**DOI:** 10.3390/tropicalmed9030055

**Published:** 2024-02-28

**Authors:** Mayumi Manamperi, P. Kandegedara, G. I. C. L. De Zoysa, J. M. A. I. K. Jayamanna, E. G. Perera, N. D. Asha Dilrukshi Wijegunawardana

**Affiliations:** 1Ministry of Health, Nutrition, and Indigenous Sciences, Colombo 10, Sri Lanka; mayumimadhushani83@gmail.com; 2Postgraduate Unit, Faculty of Applied Sciences, Rajarata University of Sri Lanka, Mihintale 50300, Sri Lanka; 3Department of Health Promotion, Faculty of Applied Sciences, Rajarata University of Sri Lanka, Mihintale 50300, Sri Lanka; prabuddhika@as.rjt.ac.lk; 4Department of Electric and Electronic Technology, Faculty of Technology, Rajarata University of Sri Lanka, Mihintale 50300, Sri Lanka; isuruchamara8@gmail.com; 5Department of Bioprocess Technology, Faculty of Technology, Rajarata University of Sri Lanka, Mihintale 50300, Sri Lanka; isharaikj@gmail.com (J.M.A.I.K.J.); geethanjana91@yahoo.com (E.G.P.)

**Keywords:** awareness, KAP survey, leishmaniasis, questionnaire, socio-economic factors

## Abstract

Leishmaniasis is one of the neglected tropical diseases. Studies show that the poor knowledge about epidemiological aspects of leishmaniasis within communities causes the collapse of existing disease control programs. Therefore, the present study focuses on a detailed survey of the existing awareness among the threatened population in the Medawachchiya Public Health Inspector’s (PHI) Area in the Anuradhapura District, Sri Lanka, aiming to assist the health staff to organize community-based vector control programs effectively in the future. Assessment of the awareness of residents of two hundred and seventy households (*n* = 270) from 10 Grama Niladhari Divisions (GNDs) was carried out by using a structured questionnaire. Among 143 females and 134 males, only 75.1% had knowledge about the disease, 5.8% (*n* = 16) of the participants knew only about the vector, and 28.9% (*n* = 80) knew about control methods. The study showed a considerable lack of awareness about the disease among the studied population. The study found that age and education levels had significant impacts on knowledge, attitudes, and practices. However, factors like gender, marital status, occupation, income, and expenses did not show significant correlations. The present study suggests huge scope for greater achievements in community-related vector control methods by implementing a continuous educational program.

## 1. Background

### Why a KAP Survey?

Leishmaniasis, one of the neglected tropical diseases, has been reported to infect people in regions of Asia, Africa, the Middle East, and the central and southern parts of the Americas [1]. It is a zoonotic disease transmitted by phlebotomine female sandflies carrying protozoan parasites of the genus Leishmania. Leishmaniasis infection can manifest itself in various clinical forms. The three primary disease phenotypes can be considered cutaneous (CL), mucosal (ML), and visceral leishmaniasis (VL) [2].

Regarding the current global scenario, the estimates for the annual occurrence of new cases of cutaneous leishmaniasis fluctuate between approximately 700,000 and 1.2 million or even higher [3]. This disease predominantly affects individuals who have experienced periods of hunger, resettlement, poor living conditions, compromised health, or face financial constraints [4].

Cutaneous leishmaniasis, caused by parasites transmitted through the bites of infected sandflies, disproportionately affects individuals facing various socioeconomic challenges. Factors such as inadequate housing, limited access to healthcare, and nutritional deficiencies significantly contribute to the prevalence of this disease. The variation in reported cases highlights the complexity and scale of this health issue, underscoring the vulnerability of populations under specific adverse conditions.

The primary clinical form of leishmaniasis in Sri Lanka is cutaneous leishmaniasis [5,6]. In 1992, the first case of locally acquired cutaneous leishmaniasis was reported in the Southern Province of Sri Lanka [7]. Since then, the disease has become endemic across the country. Anuradhapura and Hambantota are considered to be the high-risk districts in the country [8]. Certain socio-economic factors such as occupation, life style and poor knowledge about different epidemiological aspects of leishmaniasis in rural areas are associated with the incidence.

Current control measures require a successful early detection system [9]. Since there are no vaccines and medications available to prevent the infection, the best strategy to mitigate this condition is vector control and safeguards against sandfly bites. The prevention of this disease depends heavily on community involvement. In order to achieve that, community members need to be equipped with sufficient knowledge about the disease, its vector, modes of transmission, and other risk factors.

The purpose of this questionnaire survey was to evaluate the understanding, attitudes, and behaviors of the general public regarding cutaneous leishmaniasis and its different dimensions. The survey specifically took place in 10 GNDs within the Medawachchiya PHI Area in the Anuradhapura District of Sri Lanka. Its aim was to support the health sector in effectively strategizing community-based efforts for controlling disease-carrying vectors in the future. In addition to gathering basic personal information and history, the questionnaire included inquiries about individuals’ knowledge of leishmaniasis, the vectors responsible, and the preventive measures they have employed to manage or prevent the disease. The intention is to gain insights that can assist in planning and implementing more targeted and informed public health initiatives.

## 2. Methods

### 2.1. How Did It Go?

#### 2.1.1. Selection of the Study Setting

A cross-sectional study was conducted in 10 Grama Niladhari Divisions (GNDs) within the Medawachchiya Public Health Inspector (PHI) area. This area is considered to be a newly emerging high-risk zone located in the Anuradhapura district of Sri Lanka, as depicted in Figure 1. The study aimed to encompass a total target population of 300 households, with each GND representing around 30 households. These households were selected to provide a comprehensive overview of the area’s health and demographic characteristics within this specific region.

#### 2.1.2. Questionnaire Validation

A panel of three health care specialists extensively examined the questionnaire. Trainee public health specialists, two physicians (consultant parasitologist and consultant public health community physician), two entomologists, and two lecturers from the research institution covering both medical entomology and health promotion specialization were included into the panel. This content validation was performed to ensure that the questions were not unclear and that the material was suitable. Modifications to the layout and structure of questions were made in response to panel reviews. Pilot research was also conducted among randomly selected volunteer community members living in close proximity to the study location in Medawachchiya PHI area (*n* = 40) to establish dependability. The test-retest approach was used. After a 10-day delay, the same set of participants was asked to complete the same questionnaire.

#### 2.1.3. Data Collection

The random sampling method were used to collect the data from the residents in the area [10]. The data collection was carried out by interviewer administrated survey using structured questionnaire. At least one Respondent from one house was participated to the questionnaire study. Sample size was determined according to the results of pilot survey.

The questionnaire contained 38 (thirty-eight) short and pointed questions directed towards the socio-economic background, knowledge, attitudes and practices of the respondents regarding leishmaniasis. Interviewers were also carefully selected and trained prior to the survey to prevent interviewer biases (Figure 2). The area covered during the survey constituted 13.2% of the Medawachchiya MOH area.

KAP questions were scored as follows. For the knowledge part, the lowest score of 1 was given for the “Don’t know” or “No Idea” responses, while the highest number, 2, was given for “Yes” responses. For the practice questions, the lowest score given for “No” responses was 1, while the highest score given for the “Yes” responses was 2. During the attitude test, the lowest score was given for the “Definitely dissatisfied” response, and the highest score was given for the “Definitely satisfied” response. The scores in the attitude test were as follows: “Definitely dissatisfied” = 1, “Moderately dissatisfied” = 2, “Neutral” = 3, “Moderately satisfied” = 4, and “Definitely satisfied” = 5.

#### 2.1.4. Data Processing and Analysis

Questionnaire data were collected by the interviewers using Epicollect5 mobile application [11] and Google Forms. Each completed form was manually checked before extracting data to MS Excel 2017 to increase the quality of the data. The extracted data were analyzed by using Statistical Package for Social Sciences 25 (SPSS V.25). The Kolmogorov–Smirnov test was used to determine the normality of the data [12]. To show findings, descriptive and inferential statistics such as the chi-square test, Mann–Whitney U test, Kruskal–Wallis H test, and binary regression were utilized [13]. A *p*-value of less than 0.05 was considered statistically significant for each test.

#### 2.1.5. Ethical Consideration

All participants were briefed about the study prior to the survey and gave verbal consent. An anonymous questionnaire was conducted. Ethical approval for the study was given by the Ethical Review Committee of the Faculty of Applied Sciences, Rajarata University of Sri Lanka (ERC approval Ref: ERC/04/021).

## 3. Results

### 3.1. Whose Is This about?

The data presented in Table 1 provide a comprehensive overview of the participants involved in the study, offering detailed insights into various demographic parameters. These parameters encompass the distribution of participants across age categories, gender, marital status, educational attainment, occupational statuses, as well as family income and expenditure brackets. The age range of the participants spanned from 17 to 80 years, demonstrating a wide spectrum within the sample group. The average age of the participants was calculated at 45.64 years, with a standard deviation of 13.81 years, indicating the diversity in ages within the study population. Regarding ethnic representation, the survey did not explicitly account for ethnicity, primarily because approximately 90% of the study’s population belonged to a singular ethnic group. This particular homogeneity of the participants was noted but not directly examined in the survey results. The study’s findings highlighted a significant prevalence of high unemployment rates and relatively lower educational levels among the participants. It was speculated that these observations might be influenced by potential time lags within the survey methodology, suggesting that the reported data on unemployment and educational levels could be affected by temporal factors. This insight into time lags implied that the high unemployment and lower education levels may not accurately reflect the current status of the participants, but rather a result influenced by the timing of data collection.

### 3.2. What the KAP Survey Revealed?

The questionnaire’s reliability results indicated no statistically significant difference between the outcomes of both pilot studies, supported by a Cronbach’s alpha of 0.86. In terms of specific domains, the knowledge section displayed the highest Cronbach’s alpha, with the attitude and practices sections following in sequence. The comparatively lower Cronbach’s alpha in the attitude and practice domains could potentially be linked to their smaller number of questions in comparison to the knowledge section. A targeted total of 300 surveys was intended to be conducted, resulting in 277 completions, reflecting a response rate of 92%. This notably high response rate might be ascribed to direct face-to-face interaction between the researcher and the survey participants. It is worth noting that individuals who did not respond were not contacted.

#### 3.2.1. Knowledge: How Much Do They Know?

##### What Do They Know about the Disease?

The research revealed an intriguing insight into the local community’s awareness regarding both the disease and the vector responsible for its transmission. Despite 75.1% of the residents being familiar with the disease, the depth of knowledge regarding it remained relatively low. A striking 79.1% of respondents admitted to having no certainty about whether the disease is treatable or not, with a small 2.9% believing it to be incurable. In stark contrast, a mere 18.1% of the surveyed individuals were aware that the disease is, in fact, curable. Moreover, the study delved into participants’ perspectives on the symptoms associated with the disease, revealing a diversity of responses, as depicted in Figure 3. This underscores a significant knowledge gap and variation in understanding among the residents concerning the disease’s symptoms.

##### Do They Know the Vector of Leishmaniasis?

Among the respondents, 87.7% had heard of the sandfly, but only 0.4% knew that the sandfly is the vector of leishmaniasis, and 27.8% believed that sandflies bite only humans. Figure 4 and Figure 5 illustrate the participants’ knowledge about the symptoms of the disease and the breeding sites of the sandfly vector.

#### 3.2.2. Attitudes: What Do They Think?

##### Attitudes towards Vector Control, Disease Control and Knowledge Sources

The majority of participants (87.0%) primarily acquired their information from Medical Officers of Health (MOH Office) and social media platforms. Interestingly, a substantial percentage (93.0%) of these participants express dissatisfaction with their understanding of the disease and its vectors. Moreover, 82.1% of respondents were dissatisfied with the knowledge levels of their family members regarding the subject. Although 85.9% of the respondents gathered knowledge from community members, a significant portion (59.55%) expressed dissatisfaction with the information provided by these community sources. Notably, this dissatisfaction indicated a gap in knowledge sharing within the community.

Through these findings, participants highlighted the necessity for increased involvement by both governmental and non-governmental health organizations in disseminating accurate information. This call suggests a need for these entities to play a more active role in educating the public and raising awareness about the disease and its vectors.

### 3.3. Practices: Are They Successful?

#### 3.3.1. What Is Their Level of Vector Control Practices?

The research findings revealed that within the studied population, there is a notable prevalence of common vector control practices. This high prevalence was largely attributed to various educational initiatives led by entities such as the National Dengue Control Unit, anti-malaria campaigns, health office physicians, as well as both governmental and non-governmental health organizations. Despite this, a portion of respondents (16.1%) acknowledged the significance of education on vectors. However, a substantial majority (93.0%) expressed dissatisfaction with their current level of knowledge regarding vectors and the associated diseases. Furthermore, the majority (64.7%) identified the inability to recognize vectors and their breeding sites as a primary hindrance to effectively practicing vector control. This obstacle is considered serious in terms of controlling not only the vector studied but also those of other diseases, like dengue and malaria, which are transmitted through similar vectors.

#### 3.3.2. What about the Level of Disease Control Practices?

In the study, a substantial majority of respondents (99.6%) expressed their lack of understanding regarding the appropriate actions to take when dealing with individuals afflicted by this disease. Many indicated a significant knowledge gap, stating they were uncertain about the symptoms and often confused between typical skin conditions like pimples or blisters and the specific symptoms of leishmaniasis. Furthermore, the study revealed a concerning trend: the lower the number of reported deaths attributed to this disease, the less emphasis and attention it receives. This lack of attention contributes to a general unawareness of how to effectively treat patients suffering from leishmaniasis.

#### 3.3.3. What Does the Inferential Study Say?

The descriptive statistics for each item of the questionnaire are presented in Table 2. In the knowledge domain, the majority, 212 (71%), could not correctly identify leishmaniasis as a disease that causes sores and scars, while 130 (46.75%) participants knew the disease vector sandfly. A majority of 226 (81.68%) had no idea of the risk factors for leishmaniasis disease, while 48 (17.44%) responded that they had some idea of the risk factors associated with transmission of leishmaniasis disease. Among the risk factors, 276 (99.6%) participants could not identify any possible reservoir hosts for the leishmaniasis parasite, followed by 165 (59.60%) who were unable to identify sandfly breeding sites, and 145 (52.30%) stinging time, as risk factors for transmission of leishmaniasis. A majority of 229 (82.60%) did not know about the control measures for sandflies or how to prevent the transmission of the disease. In the attitudes section, 165 (59.40%) stated that attitudes towards sandfly vector control were poor and not known, while a majority of 212 (48%) agreed that attitudes towards disease control were poor in their family and community. Similarly negative attitudes were observed in the responses to questions about information on leishmaniasis and its control (223, 80.43%). The frequency of responses to questions about practices showed that the majority of participants (276, 99.6%) rarely thought about the precautions that should be taken to avoid transmission of the disease by infected patients. The majority indicated that disease and sandfly vector control practices by the government (253, 91.3%), private institutions (244, 88.1%), community members (238, 85.9%), family members (133, 48.0%), and themselves (259, 93.5%) were inadequate. A large number mentioned community members (143, 51.6%), followed by television (52, 18.8%) as sources of information, while only 20 (7.2%) mentioned health professionals as sources of information.

The Mann–Whitney U and Kruskal–Wallis H tests were applied to compare the values of each area with different demographic factors. The results of the test are shown in Table 3. Considering *p* < 0.05 as statistically significant, there was a difference between the mean knowledge (*p* = 0.037), attitude (*p* = 0.044), and practices (*p* = 0.041) scores of the different age categories (Table 3). The mean knowledge scores of the different educational categories also differed significantly (*p* = 0.021), with higher levels of education (up to 9th grade) having higher knowledge than lower levels of education and no schooling. However, there were no higher educated participants from the same group of study participants representing other educational categories, such as above 9th grade to college degree. The number of family members also showed that more than five members had statistically significant higher mean scores for attitude and practice than the other two categories, i.e., the number of family members of less than two and between two and five (Table 3). No statistically significant differences were found in the results for other demographic factors such as gender, marital status, occupational status, monthly income, and expenses (Table 3).

A correlation between the different areas of the questionnaire was also examined. A positive correlation was found between knowledge and attitude and between attitude and practice (Table 4). Using the chi-square test, the study participants’ scores for each area were categorized into three categories: good, fair, and poor. The results of this categorization are shown in Table 5.

## 4. Discussion

The current study differs from previously conducted KAP studies in several aspects. First, the current study included participants of different age groups (19 to 85 years). Second, a scoring system was developed, and participants’ scores for each domain were analyzed and related to various demographic factors; participants were also categorized according to their scores for each domain. Another important aspect of this study was to evaluate different attitudes and practices of the low educated population regarding leishmaniasis and to analyze different practices that may serve as risk factors for transmission of leishmaniasis parasites and acquisition of the disease. The results of our study show that knowledge of leishmaniasis disease transmission and sandfly vector control and attitudes and practices related to these two aspects were higher in older people than in young people within the study participants, residents of the Medawachchiya PHI area. The same results were also observed with a slightly higher level of education in the study population and when there were more than five family members within the family (Table 3 and Table 4).

We should commonly admit that so-called neglected diseases such as leishmaniasis have exclusively affected extremely poor populations living in remote areas beyond the reach of health services [14]. As it is rarely fatal and most commonly found in poor communities, there is little or no attention regarding this disease from researchers, drug developers and healthcare authorities. To our knowledge, this is the first time ever that a KAP survey has been carried out in this area regarding this matter.

The impact from years of awareness-raising and education campaigns by several governmental and non-governmental organizations such as the National Dengue Control Unit, Anti Malaria Campaign, MOH offices, etc., have increased common vector control practices such as cleaning mosquito breeding sites and using personal protection measures, insect repellents and attractants. But sandfly-specific vector control practices are still new to the studied population.

Early diagnosis is a key factor in the management of the disease. In Sri Lanka, no programs are conducted to identify patients in the early stages of infection [15]. This has resulted in a lack of knowledge among the population. In addition to that, late identification causes more complicated situations. Therefore, the study clearly indicated that there is a need for a series of education programs.

In the present study, the main knowledge resources among the respondents were social media, MOH offices, and other community members. Even though a knowledge gap was observed among the respondents in terms of understanding the vector, vector control practices, early identification of the disease and seeking medical attention, there is a huge scope for community-directed vector and disease control programs like the earlier community-based programs for Anopheline eradication from the rice ecosystem [16].

Although selection was unbiased, out of the total number of respondents (*n* = 277), more than half (*n* = 143) were female. Most of them were housewives within the age range of 34–49 with an ordinary level of education. We think this did not affect the results of the study, as it is natural for housewives or mothers to be the key persons that lead the family in vector control practices of this kind and in recognizing early symptoms of disease. This indicates that younger housewives and mothers must be considered as key elements for successful disease control programs [17]. However, the maintenance and removal of sites such as rock cervical, termite nests, and animal burrows in areas outside the immediate living area are mainly taken care of by males. Therefore, vector control programs should target males and females separately, with attention to the specifics of their gender roles in the households [18,19].

The majority of respondents do not seek medical care after the identification of leishmaniasis disease symptoms; they pointed out that people tend to seek traditional remedies, such as topical administration of honey or some herbal extracts, first because of various socio-economic factors such as lower economic background and relatively poor job status. Most of the respondents were self-employed like farmers and hired workers. Even though most of the respondents had monthly income >LKR 50,000, monthly expenses exceeded their income. In addition to that, all nearby dermatological clinics are more than 20 km away, requiring a day to be spent seeking medical attention.

This kind of situation is common in other leishmaniasis-endemic countries such as North Sudan and Bolivia [11,12]. Although this study did not focus on this matter, it is important to study further whether not seeking medical attention right after the identification of the symptoms and tending towards traditional remedies have a direct relationship with economic background and job status.

One of the major limitations in our study was the time period; also, the study population could not be extrapolated to the whole country. However, this study will be a basis for future studies and successful vector and disease control programs. The results of this KAP survey also will help achieve our target of improving early diagnosis and community-directed vector control programs not only in the study area but ultimately across the entire nation.

### What Are the Challenges? Limitations of the Research

As mentioned earlier, the researchers employed a convenience sampling method to collect survey responses, where individuals who did not respond were not pursued for further participation. Consequently, a more robust research design using random stratified sampling could have been considered. The use of such a method would involve dividing the population into distinct subgroups or strata and then selecting participants randomly from each group, potentially offering a more representative sample. Given that the research participants were exclusively drawn from a singular community within the study area, the sample may not entirely represent the entire demographic diversity of the resident population. This limitation could impact the generalizability of the study findings beyond that specific community.

## 5. Conclusions

The current research underscores the crucial significance of initiating a range of community-oriented awareness initiatives. These include health camps, community meetings, and the dissemination of information through various media channels, particularly leveraging the outreach potential of social media platforms. Such strategic endeavors prove to be highly effective in reshaping perspectives toward early diagnosis, encouraging prompt medical attention, and effectively managing disease vectors. Furthermore, drawing from the insights gained from past experiences in handling infectious diseases, a portion of these lessons should be allocated and adapted to the specific disease and vector control programs at hand.

Our study’s findings emphasize the pressing necessity for tailored educational and awareness campaigns, specifically targeting university students across various age brackets. Designing educational programs rooted in the four key elements of the health belief model is recommended. These programs should be not only meticulously devised but also implemented on a substantial scale to yield maximum impact and reach.

## Figures and Tables

**Figure 1 tropicalmed-09-00055-f001:**
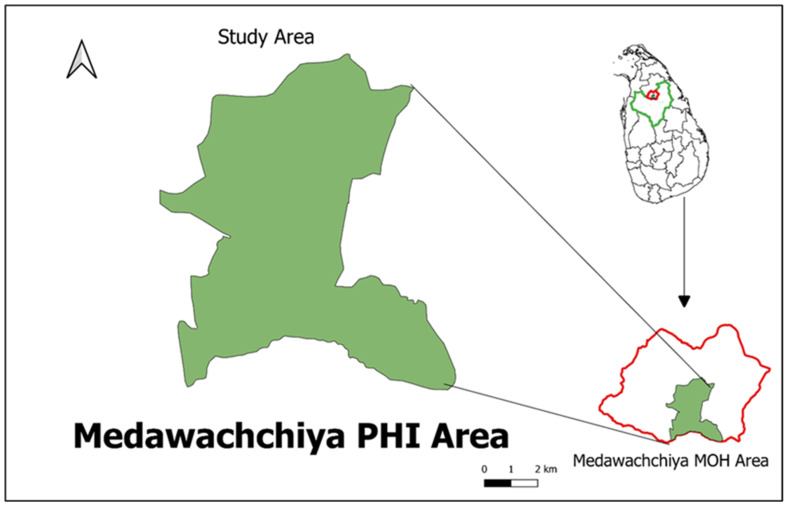
Study area: Medawachchiya PHI area, Anuradhapura District, Sri Lanka.

**Figure 2 tropicalmed-09-00055-f002:**
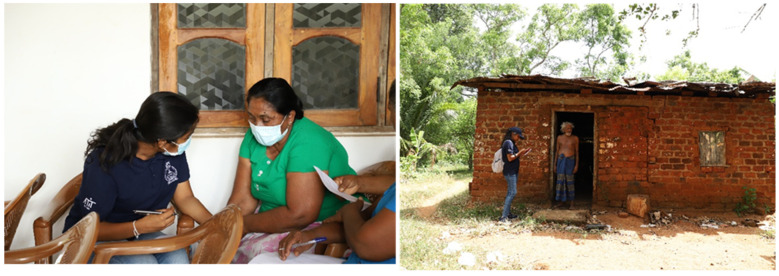
Interviewer-administered questionnaire interviews conducted in the study area.

**Figure 3 tropicalmed-09-00055-f003:**
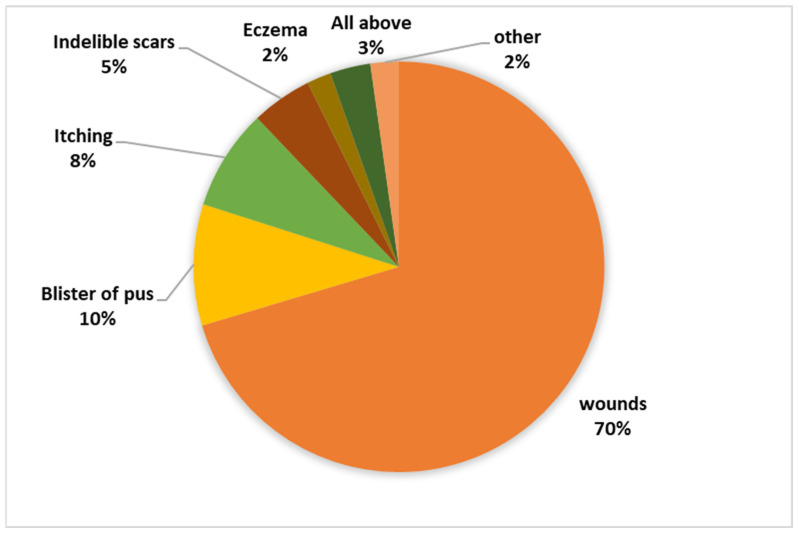
Knowledge regarding symptoms associated with the disease.

**Figure 4 tropicalmed-09-00055-f004:**
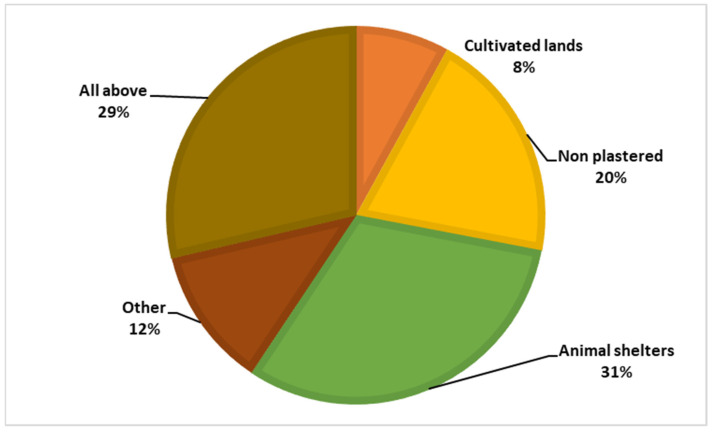
Knowledge regarding the vector’s breeding sites.

**Figure 5 tropicalmed-09-00055-f005:**
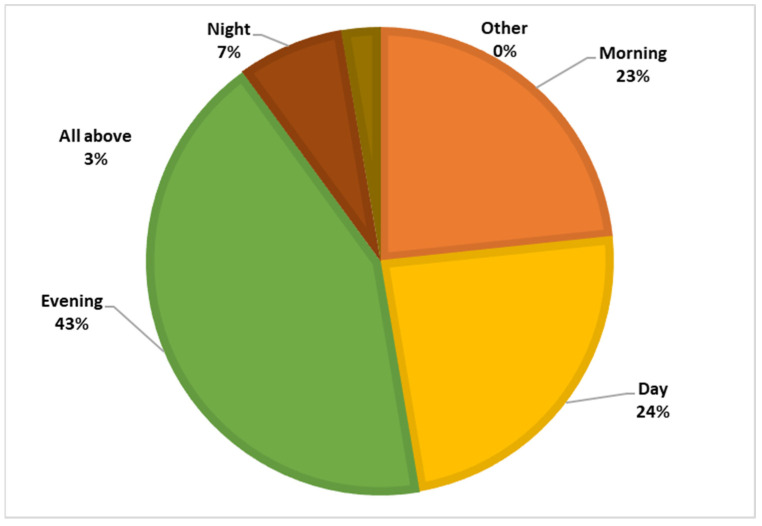
Knowledge regarding the biting period.

**Table 1 tropicalmed-09-00055-t001:** Demographic profile of the study participants.

Variable	N (%)	Variable	N (%)
Age (Mean ± SD)	45.64 ± 13.81	Education	
<18 years	2 (0.7%)	No schooling	174 (62.8%)
18–33 years	48 (17.3%)	Up to Grade 5	84 (30.3%)
34–49 years	126 (45.5%)	Up to Grade 9	19 (6.9%)
50–65 years	73 (26.4%)		
>65 years	28 (10.1%)	Monthly income	
Gender		<LKR 20,000	61 (22%)
Male	134 (48.4%)	LKR 200,000–40,000	118 (42.6%)
Female	143 (51.6%)	>LKR 40,000	98 (35.4%)
Marital status			
Single	25 (9%)	Monthly expenses	
Married	252 (91%)	<LKR 20,000	36 (13%)
Occupational status		LKR 200,000–40,000	84 (30.3%)
Unemployed	105 (37.9%)	>LKR 40,000	157 (56.7%)
Self-employed	77 (27.8%)	Number of family members	
Employed	95 (34.3%)	<2	25.0 (9%)
Government sector	70 (25.3%)	2–4	183 (66.1%)
Private sector	25 (9.0%)	>5	69 (24.9%)

**Table 2 tropicalmed-09-00055-t002:** Frequency of various responses to questions.

Domain	* Item No.	Yes	No	Don’t Know
Knowledge				
Disease symptoms/Complications	1	68 (75.1%)	208 (24.5%)	1 (0.4%)
	2	221 (79.8%)	56 (20.2%)	-
	2a	30 (10.8%)	247 (89.2%)	-
	2b	25 (9.0%)	252 (91.0%)	-
	2c	15 (5.4%)	262 (94.6%)	-
	2d	6 (2.2%)	271 (97.8%)	-
	2e	10 (3.6%)	267 (96.4%)	-
	2f	7 (2.5%)	270 (97.5%)	-
	2g	3 (1.1%)	274 (98.9%)	-
	3	50 (18.1%)	8 (2.9%)	219 (79.1%)
Vector	4	16 (5.8%)	235 (84.8%)	26 (9.4%)
	5	243 (87.7%)	12 (4.3%)	22 (7.9%)
Risk factors	6	77 (27.8%)	108 (39.0%)	91 (32.9%)
	7	151 (54.5%)	126 (45.5%)	-
	7a	90 (32.5%)	187 (67.5%)	-
	7b	44 (15.9%)	233 (84.1%)	-
	7c	10 (3.6%)	267 (96.4%)	-
	7d	1 (0.4%)	276 (99.6%)	-
	7e	1 (0.4%)	276 (99.6%)	-
	8	112 (40.4%)	165 (59.6%)	-
	8a	12 (4.3%)	265 (95.7%)	-
	8b	30 (10.8%)	247 (89.2%)	-
	8c	47 (17.0%)	230 (83.0%)	-
	8d	18 (6.5%)	259 (93.5%)	-
	8e	43 (15.5%)	230 (83.0%)	4 (1.4%)
	10	132 (47.7%)	145 (52.3%)	-
	10a	35 (12.6%)	241 (87.0%)	1 (0.4%)
	10b	36 (13.0%)	241 (87.0%)	-
	10c	64 (23.1%)	213 (76.9%)	-
	10d	11 (4.0%)	266 (96.0%)	-
	10e	4 (1.4%)	273 (98.6%)	-
	10f	-	277 (100%)	-
Control measures	11	80 (28.9%)	197 (71.1%)	-
	11a	88 (31.8%)	189 (68.2%)	-
	11b	85 (30.7%)	192 (69.3%)	-
	11c	30 (10.8%)	247 (89.2%)	-
	11d	10 (3.6%)	267 (96.4%)	-
	11e	4 (1.4%)	273 (98.6%)	-
	11f	38 (13.7%)	239 (86.3%)	-
Attitude	12	75 (27.1%)	53 (19.1%)	149 (53.8%)
	13	50 (18.1%)	50 (18.1%)	180 (65.0%)
	14	46 (16.6%)	213 (76.9%)	18 (6.5%)
	15	46 (16.6%)	210(76.5%)	18 (6.5%)
	16	45 (16.2%)	209 (75.8%)	22 (7.9%)
	17	3 (1.1%)	253 (91.3%)	21 (7.6%)
	18	7 (2.5%)	244 (88.1%)	26 (9.4%)
	19	1 (0.4%)	238 (85.9%)	38 (13.7%)
	20	118 (42.6%)	133 (48.0%)	23 (8.3%)
	21	-	259 (93.5%)	18 (6.5%)
Practice	22	189 (68.2%)	88 (31.8%)	-
	23	173 (62.5%)	104 (37.5%)	-
	24	20 (7.2%)	257 (92.8%)	-
	25	14 (5.1%)	263 (94.9%)	-
	26	52 (18.8%)	225 (81.2%)	-
	27	13 (4.7%)	264 (95.3%)	-
	28	143 (51.6%)	134 (48.4%)	-
	29	5 (1.8%)	272 (98.2%)	-
	30	11 (4.0%)	266 (96.0%)	-

* Item numbers description was given as the Appendix A.

**Table 3 tropicalmed-09-00055-t003:** Mean score with respect to demographic variables.

Variable		Knowledge Score 1.82 + 0.197 Mean Rank	*p*-Value	Attitude Score 1.63 + 0.341 Mean Rank	*p*-Value	Practices Score 1.48 + 0.0341 Mean Rank	*p*-Value
Gender	Male	140.51	0.756	140.60	0.725	140.89	0.675
	Female	137.58		137.29		136.98	
Age	<18 years	199.00	0.037	157.75	0.044	157.75	0.041
	18–33 years	131.67		131.75		134.45	
	34–49 years	140.32		128.79		127.71	
	50–65 years	153.83		163.17		163.17	
	>65 years	102.70		133.00		133.00	
Marital status	Single	136.48	0.866	123.60	0.303	149.80	0.466
	Married	139.25		140.53		137.93	
Occupational status	Unemployed	134.48	0.400	139.83	0.127	139.83	0.127
	Self employed	131.30		121.91		121.91	
	Employed						
	Government sector	154.66		153.89		153.89	
	Private sector	139.78		149.42		149.42	
	Other	125.31		112.63		112.63	
Education	No schooling	129.29	0.021	132.07	0.149	132.07	0.149
	Up to Grade 5	152.72		152.08		152.08	
	Up to Grade 9	167.26		144.66		144.66	
Monthly income	<10,000	140.07	0.354	122.55	0.179	122.55	0.179
	10,000–20,000	122.42		128.98		128.98	
	20,000–30,000	133.45		137.72		137.72	
	30,000–40,000	129.90		135.91		135.91	
	40,000–50,000	151.94		170.14		170.14	
	>50,000	153.31		137.85		137.85	
Monthly expenses	<10,000	79.00	0.058	86.50	0.306	86.50	0.306
	10,000–20,000	121.30		120.40		120.40	
	20,000–30,000	125.57		141.26		141.26	
	30,000–40,000	130.62		150.61		150.61	
	40,000–50,000	158.50		133.71		133.71	
	>50,000	147.02		143.47		143.47	
Number of family members	<2	125.66	0.690	82.36	0.000	82.36	0.000
	2–4	139.81		138.46		138.46	
	>5	139.71		159.24		159.24	

*p*-value < 0.05 is statistically significant; Kruskal–Wallis H test. Note: K-score = average knowledge score; A-score = average attitude score; P-score = average practice score.

**Table 4 tropicalmed-09-00055-t004:** Correlations between knowledge, attitudes, and practices.

Variables	Spearman’s Rho	*p*-Value
Knowledge, attitude	0.099 *	0.049
Knowledge, practice	0.060	0.159 ^#^
Practice, attitude	0.358 *	0.000

* Correlation is significant at the 0.05 level (1-tailed). ^#^ Smaller rho coefficients with *p*-value of more than 0.05 indicate that there is not a statistically significant association between the two ordinal variables of knowledge and practices.

**Table 5 tropicalmed-09-00055-t005:** Categorization of study participants score on KAP domains.

Variable		Knowledge			*p*-Value	Attitude			*p*-Value	Practice			*p*-Value
		Good	Fair	Poor		Good	Fair	Poor		Good	Fair	Poor	
**Gender**	Male	130	4	0	0.810	96	38	0	0.622	32	98	4	0.803
	Female	138	5	0		101	41	1		31	106	6	
**Age**	<18 y	2	0	0	0.993	2	0	0	0.399	0	2	0	0.614
	18–33 y	47	2	0		37	12	0		16	32	1	
	34–49 y	121	4	0		82	43	0		27	92	6	
	50–65 y	71	2	0		57	15	1		13	57	3	
	>65 y	27	1	0		19	9	0		7	21	0	
**Marital status**	Single	25	0	0	0.337	14	11	0	0.193	9	15	1	0.243
	Married	243	9	0		183	68	1		54	189	9	
**Occupation status**	Unemployed	102	3	0	0.703	72	32	1	0.156	31	71	3	0.069
	Self employed	67	2	0		43	26	0		6	68	3	
	Employed												
	Government sector	68	2	0		58	12	0		16	51	3	
	Private sector	23	2	0		20	5	0		8	17	0	
	Other	8	0	0		4	4	0		2	5	1	
**Education**	No schooling	168	6	0	0.791	116	57	1	0.304	36	131	7	0.802
	Up to Grade 5	82	2	0		66	18	0		22	600	2	
	Up to Grade 9	18	1	0		15	4	0		5	13	1	
**Monthly income**	<10,000	28	1	0	0.956	13	16	0	**0.031 ***	7	19	3	0.311
	10,000–20,000	31	1	0		24	8	0		6	25	1	
	20,000–30,000	59	3	0		46	15	1		9	51	2	
	30,000–40,000	54	2	0		39	17	0		11	43	2	
	40,000–50,000	35	1	0		32	4	0		11	25	0	
	>50,000	61	1	0		43	19	0		19	41	2	
**Monthly expenses**	<10,000	6	0	0	0.641	1	5	0	0.073	2	3	1	0.150
	10,000–20,000	28	2	0		20	10	0		8	20	2	
	20,000–30,000	36	2	0		26	11	1		8	29	1	
	30,000–40,000	46	0	0		36	10	0		9	33	4	
	40,000–50,000	45	1	0		35	11	0		15	31	0	
	>50,000	107	4	0		79	32	0		21	88	2	
**Number of family members**	<2	25	0	0	0.780	9	16	0	**0.000 ***	10	13	2	0.244
	2–4	176	7	0		131	52	0		36	141	6	
	>5	67	0	0		57	11	1		17	50	2	

* Correlation is significant at the 0.05 level (1-tailed).

## Data Availability

Data are contained within the article and Appendix A.

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
