# Peer review of "What Does a KAP Survey Reveal about the Awareness Regarding Leishmaniasis among the Community of an Endemic Area in Sri Lanka?"

_tropicalmed, 2024, doi:10.3390/tropicalmed9030055_

Round 1

Reviewer 1 Report

Comments and Suggestions for Authors

Comments on the Quality of English Language

Generally good but need some improvement

Author Response

Reviewer’s comment

Amendments

Reviewer 1

Line 17: Please define the abbreviation PHI for the first time use in the abstract

section or any other sections in the manuscript.

Line 17: Please define the abbreviation GNDs for the first time use in the abstract

section or any other sections in the manuscript.

Corrected.

Line 24: Please insert full stop after the phrase “among the studied population”

Corrected.

Keywords:

It seems too long or many keywords

Corrected.

1. Background

This section seems Okay but some sentences and paragraphs need to make careful

revisions of the grammar and English language so as to avoid readers of the manuscript.

Corrected.

2. Methods

This section seems okay.

Lines 70-110: I suggest to insertion of appropriate references where it is needed.

Address the comment.

Lines 162-165: please include high-quality of photos of the 7 species of the rodent

studied.

Nothing related to rodents here, instead respondents to the study questionnaire.

“The results of our study show that knowledge of disease

transmission and vector control, attitudes and practices related to these two aspects

were higher in older people than in young people” This needs detail explanation

Corrected.

Lines 262 -264: “The majority of the respondents do not seek medical care after

the identification of the symptoms, they pointed out that people tend to seek

traditional remedies first be-263 cause of various socio-economic factors such

economical background, job status.” This requires detail explanations.

Corrected.

Table 4: Authors should provide a detailed explanation for the absence of correlation

between Knowledge & practice.

Corrected.

References

Authors need to avoid old references

Corrected where appropriate.

Need to check extensive revision and double-check grammar and phrasing

Corrected as much as possible.

How was the sample eventually defined?

Corrected.

How were the participants recruited?

Corrected.

Comments on Tables

Corrected.

Comments on the results section

Corrected.

Comments on multivariable analysis

Corrected.

Reviewer 2 Report

Comments and Suggestions for Authors

Estimated Authors, thank you for the opportunity to review this interesting paper from Wjegunawardana et al. 

in this very interesting paper from Sri Lanka, Authors address the topic of KAP of residents from a region of Sri Lanka about Leishmaniasis. Eventual results are quite unsatisfying. Respondents exhibited a very low awareness of the addressed topic. Moreover, no effective predictor variables for the knowledge status were eventually identified.

From my point of view, the present article could deserve acceptance and eventual publication, but several amendments and improvements are required.

To begin with, the overall quality of the English text is far from optimal. As non English speaker, I'm never comfortable in stressing the quality of the English text from other Authors, but in this case I cannot avoid the request for extensive revision and double check of grammar and phrasing.

Second, Authors should describe in more details the following topica:

- how was the sample eventually defined? was a preventive sample size calculation performed or it was a convenience sampling? Please explain. The sentence "A total of 300 surveys were targeted to conducted, with 277 completions, for a 92% response rate" is unclear.

- how were participants recruited? You write that: "The random sampling method were used to collect the data from the residents. The data collection was carried out by interviewer administrated survey using structured questionnaire". Please provide further details about the selection procedures.

- Table 1 should be re-designed in order to shift from a 4 columns design to a 2 columns one.

- Table 2 is scarcely understandable without a more appropriate explanation of the included items.

- Table 3 is unclear. If I guess correctly, the values in the headers are the average +/- SD. Please remove this information, as it would be confusing. 

- Results section: please shift to a more conventional approach. Describe in full details the demographics of your sample, then move to a summary of knowledge, attitudes, and practices reported.

Finally, please take into account that your study did not include a multivariable analysis. As a consequence, you can only discuss about a descriptive analysis, while any inference about the causative relationships between assessed variables should be avoided.

Comments on the Quality of English Language

The overall quality of the English is inappropriate for acceptance. Extensive revision is warranted.

Author Response

Dear Sir/Madam,

Thank you for reviewing the above article and again for the valuable comments. We addressed all those comments where appropriate and included the suggested changes accordingly. Please find the attached document for your kind reference.

Thank you and kind regards,

Asha Wijegunawardana

Round 2

Reviewer 2 Report

Comments and Suggestions for Authors

Estimated Authors,

I'm congratulating with you for the high quality editing you've performed on your paper. The paper has been revised according to my previous review. I've no further requests, and I'm endorsing its eventual acceptance.

Comments on the Quality of English Language

Some minor typos that could be fixed by Authors during publication stages.